# *Sarracenia alata* (Alph.Wood) Alph.Wood Microcuttings as a Source of Volatiles Potentially Responsible for Insects’ Respond

**DOI:** 10.3390/molecules26092406

**Published:** 2021-04-21

**Authors:** Jacek Łyczko, Jacek Piotr Twardowski, Bartłomiej Skalny, Renata Galek, Antoni Szumny, Iwona Gruss, Dariusz Piesik, Sebastian Sendel

**Affiliations:** 1Department of Chemistry, Wrocław University of Environmental and Life Sciences, ul. Norwida 25, 50-375 Wrocław, Poland; antoni.szumny@upwr.edu.pl; 2Department of Plant Protection, Wrocław University of Environmental and Life Sciences, Grunwaldzki Sq. 24a, 50-363 Wroclaw, Poland; jacek.twardowski@upwr.edu.pl (J.P.T.); iwona.gruss@upwr.edu.pl (I.G.); 3Department of Genetics, Plant Breeding and Seed Production, Wrocław University of Environmental and Life Sciences, Grunwaldzki Sq. 24a, 50-363 Wrocław, Poland; bartlomiej.skalny@wp.pl (B.S.); renata.galek@upwr.edu.pl (R.G.); 4Department of Biology and Plant Protection, UTP University of Science and Technology, Al. prof. S. Kaliskiego 7, 85-796 Bydgoszcz, Poland; dariusz.piesik@utp.edu.pl (D.P.); sebastian.sendel@utp.edu.pl (S.S.)

**Keywords:** yellow trumpets, in vitro, bioassays, pyridine, HS-SPME, *Drosophila hydei*, *Acyrthosiphon pisum*

## Abstract

Rare carnivorous plants representing the genus *Sarracenia* are perceived as very interesting to scientists involved in various fields of botany, ethnobotany, entomology, phytochemistry and others. Such high interest is caused mainly by the unique capacity of *Sarracenia* spp. to attract insects. Therefore, an attempt to develop a protocol for micropropagation of the *Sarracenia alata* (Alph.Wood) Alph.Wood, commonly named yellow trumpets, and to identify the specific chemical composition of volatile compounds of this plant in vitro and ex vivo was undertaken. Thus, the chemical volatile compounds excreted by the studied plant to attract insects were recognized with the application of the headspace solid-phase microextraction (HS-SPME) coupled with the GC-MS technique. As the major volatile compounds (*Z*)-3-hexen-1-ol (16.48% ± 0.31), (*E*)-3-hexen-1-ol acetate (19.99% ± 0.01) and β-caryophyllene (11.30% ± 0.27) were identified. Further, both the chemical assumed to be responsible for attracting insects, i.e., pyridine (3.10% ± 0.07), and whole plants were used in in vivo bioassays with two insect species, namely *Drosophila hydei* and *Acyrthosiphon pisum*. The obtained results bring a new perspective on the possibilities of cultivating rare carnivorous plants in vitro since they are regarded as a valuable source of bioactive volatile compounds, as including ones with repellent or attractant activity.

## 1. Introduction

The genus *Sarracenia* (ca. 11 species) belongs to the family *Sarraceniaceae*, which also contains the closely allied genera *Darlingtonia* Torr. (monotypic) and *Heliamphora* Benth. (ca. 23 species) [1]. The phylogenetic relationships of this carnivorous plant genus have recently been elucidated using target enrichment. These pitcher plants are found in wet swampy areas with a lot of sunlight. *Sarracenia* are perennial-like plants which wither in winter and sprout again in springtime, blooming every year. The more light the plant receives, the stronger its colour [2,3].

Habitat destruction and over-harvesting by hobbyist have made insectivorous plants rare, including *S. oreophila* (Kearney) Wherry as well as *S. leucophylla* Raf. and *S. purpurea* L. The seeds of the genus representatives have limited dispersal and low rates of establishment. *Sarracenia alata* (Alph.Wood) Alph.Wood [4,5] are habitat specialists with a patchy distribution [6], confined primarily to longleaf pine savannahs characterized by abundant sunlight and frequent fires, and their status of conservation is NT (“Near Threatened”). The plant is a vulnerable species entered into the IUCN (International Union for Conservation of Nature) Red List of Threatened Species [7]. Carnivorous plants have become an important ornamental component of botanical garden collections. They are highly valued as ornamental plants, and the constant appearance of new varieties proves the demand of the market [8]. *S. purpurea* and *S. flava* have been the most studied in terms of metabolite profile and medical usefulness [9] because they were traditionally used by many aboriginal communities as medicinal plants against a wide spectrum of diseases, including tuberculosis. These facts as well as the low propagation rate of *Sarracenia* spp. in their natural environment are the reason for treating these plants as an alternative source of propagation—in vitro reproduction. Since, additionally, many of these plants are endangered, the development of micropropagation techniques is a very good alternative to the acquisition of secondary metabolites. In vitro cultures also provide a possibility of deriving lines from individual seedlings, which is important for stabilizing the biological material, taking into account high pollination and the ease of creating hybrids. The derivation of clones from individual seeds is crucial for the selection of compensated biological material, which may be a valuable source of secondary metabolites and new varieties. The knowledge of how to obtain microcuttings is limited [10] as micropropagation is carried out mainly in private companies.

Nowadays, the great interest of scientists is raised by the role of plants secondary metabolites acting as attractants and the plants usefulness in medicine, especially regarding the bioavailability of secondary metabolites [11,12]. Various compounds in *Sarracenia* have been reported, including volatiles, flavonoids, phytochemicals and pitcher fluid composition; however, their chemical compounds are relatively poorly described in terms of particular constituents. Volatiles for instance were identified in order to recognize the similarities among different species of the *Sarraceniaceae* phylogeny [13].

Additionally, simultaneously with compound identification, the possibilities of determining and improving bioavailability of bioactive plants constituents, should be considered. Recent reports [14] focus on the relationship between compound bioavailability and compounds chemical structure, plant matrix and possible interactions which those provokes. Nevertheless, the possibilities to enhance the bioavailability of compounds such as flavonoids are possible in various models, which may include connecting flavonoid administration with mixtures (e.g., borneol/methanol) which improves compounds solubility or, on the other site, stabilizing the flavonoids structure by capping their hydroxyl groups with methyl groups [15]. The evaluation of the efficiency of these improvements is done by in vitro and in vivo models. Regarding in vitro methods, mainly, a gastrointestinal model, and the Caco-2 models are considered, while in the case of in vivo, mostly rodent models are used [16].

Other studies were conducted to define the role of chemicals as attractants for insects. In the natural environment, differences may occur in the plants’ effectiveness to trap insects depending on their classification to a taxonomic group. Gaume et al. [17] found that pitchers captured a lower proportion of flying insects when traps were closer to the ground and that plants with a larger trap width could naturally catch larger insects. A vast majority of insect visitors are known to be potential prey for *Sarracenia* plants. The visitors come from various systematic groups (ants, flies, grasshoppers, bees, wasps, moths, beetles and leafhoppers; there are also spiders among them). It is believed, however, that not only the design of the trap determines the effectiveness of the trapping. The volatiles secreted by the plant probably play a role as well [18]. Yet, the information pertaining to this topic is scarce. It is also important to understand which volatiles are responsible for attracting insects. Insects’ response to plant volatiles can be determined using the olfactometric method. The method has successfully been used for many insect groups like herbivorous beetles [19], moths [20] or flies [21]. Studies on the chemical composition in *Sarraceniaceae* concern plants from in vivo conditions [13] as opposed to other family, e.g., *Droseraceae* [22], where plants micropropagated in vitro are the main study object for numerous purposes: (1) as a very good alternative to obtaining secondary metabolites for different types of research, (2) production of cuttings on a large commercial scale and (3) micropropagation of plants for gene banks.

The aim of the current study at the first step was to elaborate the protocol for micropropagation of *Sarracenia alata* and to identify the specific chemical composition of volatile compounds of the studied plant ex vitro and in vivo. The next step was to evaluate the insect-attracting potential of *Sarracenia alata* plants in incubation and olfactometric experiments. For this purpose, flies (*Drosophila hydei*) response was tested as compared with aphids (*Acyrthosiphon pisum*). From the test species, aphids have not been described yet as potential prey for carnivorous plants.

## 2. Results

### 2.1. Efficiency of New Shoots Regeneration after I and II Passage

Two-way analysis of variance showed significant effect of interaction between the medium and cycles of passages on the following tested properties of developing *S. alata*: the height of plants, length of roots, fresh and dry weight (complete data are given in Appendix A Plants measurement). The mean values of the number of roots and dry weight did not significantly differ between passages. The summarized effects of two subcultures (after the 12th and further 14th week) on the tested medium were significant for most of the analyzed mean values of traits, except for the number leaves and length of roots (Table 1).

The new shoots and leaves proliferation was most effective on media supplemented with higher doses of BAP (6-benzylaminopurine) 1.5 up to 3.0 mg/dm^3^ in combination with 0.3 IAA (indole-3-acetic acid), and the number of new shoots ranged between 5 and 6 with 25 leaves in the first phase of experiment. Similarly, corresponding results were noticed at the second phase of experiment in the case where plants were cultivated earlier on the previously mentioned medium. On average, they produced four new shoots with 22 leaves. All subcultures were noted to develop new shoots and a root system after 12 or the next 14 weeks of culture. The best process of plant rooting was observed on the ⅓MS medium and the ⅓MS supplemented with 0.3 IAA (4 roots on average, length above 1 cm; Table 1). The media containing a higher concentration of BAP (above 1.0 mg/dm^3^) in combination with 0.3 IAA yielded the greatest fresh and dry mass of plants and the largest number of new shoots (Figure 1).

### 2.2. Chemical Composition

The volatile compounds present in *S. alata* were analyzed using the solid-phase microextraction (HS-SPME) approach on a GC-MS Varian CP-3800/Saturn 2000 apparatus. Several linear short-chain alcohols as well as aldehydes were found, e.g., 1-hexanol (2.73% ± 0.12), (Z)-3-hexen-1-ol (16.48% ± 0.31), (Z)-3-hexen-1-ol acetate (19.99% ± 0.01) (Figure 2; the more detailed chromatograms are given as Appendix A Chromatograms). The predominated sesquiterpenoid compounds were β-caryophyllene (11.30% ± 0.27) and α-bergamotene (4.15% ± 0.19). Additionally, the diterpenoid labda-8(20).12.14-triene (4.89% ± 0.09) as well as its isomers were identified (a detailed list of identified volatiles given in the Appendix A—Volatile profile of Sarracenia alata microcuttings). Nevertheless, the most surprising was the presence of pyridine in the plant aroma profile. In order to confirm that this finding is not an artifact, or an effect of contamination, a series of blank analyses was conducted (blank sample preparation and extraction process was performed like for a regular sample without plant material). Some of the compounds found in *S. alata* could play the main role as attractants of insects.

### 2.3. Bioassay with Plants

Insects’ response to *S. alata* plants was different depending on the species (a detailed description of insects behaviour given in the Appendix A—Insects behaviour in bioassays). During the first two hours aphids (*Acyrthosiphon pisum* Harris) were interested in the plants and they even made feeding attempts. At the same time flies (*Drosophila hydei* Sturtevant) moved chaotically in the test vessel. After four hours aphids were found on the bottom of the test vessel while flies were oriented more on the plant (different parts). After two days 87% of flies and 20% of aphids were found in the plant funnels (Table 2). After four days 100% of flies were digested while the number of aphids found in funnels did not change. On the other hand, 80% of aphids died due to lack of food during the incubation with *S. alata*.

### 2.4. Bioassay with Pyridine

Pyridine significantly influenced the flies’ behaviour. For the first two concentrations (0.001 and 0.01), the response was positive (0.57 and 0.23, respectively), while for the last (1%), negative response was observed (−0.14) (Figure 3). In other words, the number of attracted flies decreased with the increasing dose of the extract.

## 3. Discussion

The procedures presented in our experiment offer a valuable protocol for micropropagation of the genetic copy of selected clones of *S. alata* to be further applied for many purposes. It is very important in view of the fact that many plant species disappear from their natural environments as a result of habitat changing and collecting plants for medical or ornamental use. Our investigations showed that the new shoots and leaves proliferation was most effective on the media supplemented with higher doses of BAP (2.0 up to 3.0) in combination with 0.3 IAA or with an addition of 2.0 BAP, and the number of new shoots ranged between 6 and 8 with 25–30 leaves in the first phase of experiment. This procedure could be suitable for effective multiplication of *S. alata* for different purposes. Similar results were recorded at the second phase of experiment in the case where plants were cultivated earlier on the previously mentioned medium. On average, four new shoots with 22 leaves were obtained. All subcultures characterized root system development after 12 or next 14 weeks of culture. The best process of plant rooting was noted for the ⅓MS medium and the ⅓MS supplemented with 0.3 IAA. The medium containing a higher concentration of BAP (above 1.0 mg/dm^3^) in combination with 0.3 IAA had a significant influence and yielded the greatest fresh and dry mass of plants. Northcutt et al. [8] demonstrated that in *S. purpurea* average shoot multiplication per subculture (2 months) over two subcultures was as follows (medium; rate in parentheses): one-third strength MS without hormones (2.0), 4.6 μM kinetin (3.5), 13.3 μM BAP (5.9), and 9.1 μM *trans*-zeatin (4.8). Shoots from hormone-free and kinetin- or BAP-containing media were similar in size but those from the trans-zeatin medium were larger, sometimes double that of other treatments. In the case of three tested species, rooted plants were obtained within 3–7 weeks. In other investigations Miclea and Bernat [23] showed that ⅓ macronutrients and micronutrients at ⅓ strength of the Murashige and Skoog (MS) medium and full strength vitamins and an addition of a Plant Growth Regulator (PGR) (0.5 mg NAA, 5.0 mg BA or 0.5 mg NAA + 3.0 mg BA) were favourable for the selected *S. purpurea* plants. After 12 weeks the largest number of roots grew on the medium supplemented with 0.5 mg/dm^3^ NAA but the absence of plant growth regulators increased their length. The best shoots multiplication (3.75 new shoots) was noticed after supplementing the ⅓ MS with 5 mg/L BA. Taking into account the threat of extinction of the genus *Sarracenia,* in vitro research may have a positive impact on their future. The multiplication of large amounts of healthy material should meet the demand for capers in different research and private cultivation. In addition, the plants obtained in this way can be used to recover the damaged ecosystems.

The volatile compounds, identified through HS-SPME technique, had shown diversified profile, including short-chain alcohols, aldehydes terpenes and terpenoids. The major representatives (more than 4.00%) were pyridine (3.10% ± 0.07), (*Z*)-3-hexen-1-ol (16.48% ± 0.31), (*E*)-3-hexen-1-ol acetate (19.99% ± 0.01), *p*-cymene (4.96% ± 0.02), (*E*)-α-bergamotene (4.15% ± 0.19), β-caryophyllene (11.30% ± 0.27) and labda-8(20),13(16),14-triene (4.89% ± 0.09)., which may be found in other, various species like *Lavandula angustifolia* Mill. [24], *Coriandrum sativum* L. [25], *Mentha piperita* L. [26], *Cannabis sativa* L. [27] and *Boletus edulis* [28]. Especially interesting was high content of (*E*)-α-bergamotene, which is characteristic for *Citrus bergamia* Risso fruit [29] and pyridine, which was reported in similar amounts in *Cicer arietinum* L. [30].

The role of volatiles attracting insects, which constitute food for *Sarracenia* plants, is not well recognized. There are only suppositions that these substances are of crucial importance. According to Moran et al. [31], insects are attracted to the *Nepenthes* pitcher opening by a combination of both the visual pattern and the sweet substances produced within the pitcher. It must be further investigated whether the volatiles produced affect prey animals. Such a trial was conducted by Hotti et al. [13], who delivered a comprehensive catalogue of chemical constituents of *Sarraceniaceae* and examined the extent to which the known phylogenetic information explains the chemical composition of the plants. *Sarracenia* plants display a huge variety of unique compounds which are found only in their lid and/or pitcher. Various compounds found in *Sarracenia* have also been reported as attracting insects, including volatiles, flavonoids, phytochemicals and the pitcher fluid compositions [13].

Our research is a step further as we tried to determine how the plant as well individual chemical compounds affected the tested insects. We chose two highly different insects in order to compare their response to *Sarracenia* plants. As far as flies are concerned, other authors repeatedly report that different species are trapped by different *Sarracenia* species [18]. Therefore, we chose *Drosophila hydei*, which is described as the model for studies on the architecture of olfactory behaviour [32] and as possible prey for *Sarracenia* [33]. At the same time the possible response of aphids to carnivorous plants had not been described yet.

In the first part of the experiment flies and aphids were incubated with plants. Flies were effectively attracted to the plants funnels and digested in almost 100%, while aphids response was much weaker. Only about 20% of the latter insects were attracted and digested by *S. alata*. It is possible that aphids were trapped more accidentally than flies. The results of different field studies [18,34] confirm that Diptera are a significant constituent of the diet of carnivorous species. The specific environmental adaptation that allows plants to use animal feed is also a natural factor regulating the occurrence of certain insects. Considering the low specialization of *Sarracenia* plants, they can be used as a biological means to fight unwanted organisms. In nature, *Sarracenia* is not particularly effective in catching items of prey [18], but in closed facilities, such as greenhouses, tunnels or garden orangeries, these plants can become a solution to problems with some pests, for example certain flies from the family of the Sciaridae. Similar suggestions, regarding the possibility of using plants or their extracts as protective agents, were made for maize [35,36] or mossy sorrel [37,38].

From the compounds found in *S. alata* we chose pyridine for olfactometric study. It reveals that pyridine attracts the Mexican fruit fly [39]. Correspondingly, in other experiments pyridine compounds were used to increase trap capture of thrips [40]. To our knowledge, the experiment mentioned was the first attempt at using pyridine in olfactometric studies on insects. Our investigations show that in two first doses (0.001 and 0.01%) pyridine was an attractant to flies, while in the last concentration (1%) this compound affected repellently. To compare our results with other experiments in olfactometry, *Rhodnius prolixus* (Heteroptera) males for instance were found to prefer isobutyric acid at doses of 1 and 5 µg but not at 10 µg [41]. Furthermore, the differences in responses for *Tribolium confusum* and *Sitophilus granaries* when cereal volatiles blends were tested [42,43,44].

Based on the results obtained in other experiments we presume that pyridine among other plant volatiles is mostly responsible for attracting insects. We confirm its potential use as an insect attractant, particularly with respect to flies. Therefore, there is a need to pursue further research on this compound and its interaction with.

## 4. Materials and Methods

### 4.1. Optimization of Micropropagation of Sarracenia alata

#### 4.1.1. Induction and Stabilisation of *Sarracenia alata* In Vitro Culture

The initial material for induction of in vitro cultures were seeds of *Saracenia alata* ‘Hardin Co.’ The seeds were pre-treated with a 70% ethanol for 10 s and then disinfected in a 3% solution of sodium hypochlorite (Domestos diluted 5×) for 8 min. After three washes in sterile water, they were laid out on the ⅓MS (Murashige and Skoog) medium with addition 1 mg/dm^3^ kinetin (KIN) (Sigma Aldrich, Saint Louis, MO, USA, cat. No. K3378) and located in a refrigerator for 2 weeks. Next the culture was carried out to incubate at 20 °C, light intensity 1000 l× and photoperiod 16/8 (day/night) conditions. The seeds germinated after 3–4 weeks in in vitro condition. After stabilization of culture (28 weeks), new plants were transferred to ⅓ of MS with an addition of various combinations of growth regulators (mg/dm^3^): 1 KIN + 0.5 l IAA and 1.5 BAP + 0.3 IAA for multiplication. After obtaining the appropriate number of new plants before establishing the main experiment on optimizing micro-propagation of *Sarracenia alata* in order to eliminate the influence of the applied earlier plants growth regulators (PGRs)—two passages of plants were carried out on the medium without PGRs at four-weeks intervals.

#### 4.1.2. PGRs Influence on *Sarracenia alata* Multiplication

The effect of BAP (Sigma Aldrich, Saint Louis, MO, USA, cat. No. B3408) concentrations (0.5, 1.0, 1.5, 2.0, 2.5, 3.0 0.0 mg/dm^3^) combined with IAA (Sigma Aldrich, Saint Louis, MO, USA, cat. No. I5148) (0.3 mg/dm^3^) was tested at the first step of experiments (I passage) and second (II passage). The MS was used as the basal medium [45] with the content of macro- and micronutrients reduced to ⅓. The experiment was established in five replications, each containing six single plants (2.5 to 3.0 cm length). In the first step of the experiment, the explants acquired were measured, next cut into 2.5–3.0 cm-long sections and placed on the ⅓MS without PGRs (II passage), in the same experimental scheme as the above mentioned. They were cultivated for the subsequent 14 weeks. Such values as the length of plants [cm], number of leaves, number of new shoots, root length [cm], number of roots, plant fresh and dry weight [g] were assessed after 12 (I passage) and next 14 weeks (II passage). The cultures during I and II passage were carried out at 20 °C, light intensity of 2500 l× and photoperiod 16/8 (day/night). A two-way analysis of variance (ANOVA) was performed to evaluate the influence of PGRs on micropropagation efficiency. Additionally, one-way analysis of variance was conducted for summarized development of new shoots after two cycles of passages. Values of standard errors (SE) were calculated.

### 4.2. Chemical Analysis

The Headspace Solid-Phase Microextraction (HS-SPME) was applied to determine the volatile organic compounds present in headspace of *Sarracenia alata* cultures reared on the ⅓MS according to Łyczko et al. [46] methodology, with slight modifications. Briefly, the part of the plant immediately after separation was placed into a 5-mL headspace vial along with internal standard (phenylnaphthalene—Sigma Aldrich, Saint Louis, MO, USA, cat. No. P27402). Thereafter, the sample was conditioned on a heating plate at 70 °C for 5 min. Then, 2 cm-long Divinylbenzene/Carboxen/Polydimethylsiloxane (DVB/CAR/PDMS) SPME fibre (Supelco, Bellefonte, PA, USA) was introduced via the poli(tetrafluoroetylen) (PTFE) septum for 30 min extraction. After the extraction time, the analytes were desorbed in an apparatus injection port for 3 min at 220 °C.

As an analyzer, a “Varian CP-3800/Saturn2000” gas chromatography mass spectroscopy (GC-MS) apparatus, equipped with a Zebron ZB-5 MSI (30 m × 0.25 mm × 0.25 µm) column, was used. GC oven temperature was programmed from 50 °C to 130 °C at rate 4.0 °C/min, then to 180 °C at rate 10.0 °C/min, and finally to 280 °C at rate 20.0 °C/min. Identification of all volatile constituents was based on a comparison of the experimentally obtained mass spectra of compounds with the mass spectra available in the NIST14 database. Additionally, the experimentally obtained linear retention indices (LRIs), calculated against C_6_-C_30_
*n*-alkanes mix (Sigma-Aldrich), were compared with the LRIs available in the The National Institute of Standards and Technology (NIST) WebBook and literature data [47]. The LRI filter was set at ± 15 points and only compounds with the similarity score ≥90% were considered as correct hits.

### 4.3. Bioassay with Insects

#### 4.3.1. Bioassay with Plants

Two plants of *Sarracenia alata* were placed in a 5 L glass vessel. Plants were comparable in terms of size and number of funnel leaves. The response of insects to the exposure to *Sarracenia alata* plants was investigated. The tested groups of insects were *Drosophila hydei* (Diptera) and *Acyrthosiphon pisum* (Hemiptera). *D. hydei* specimens were obtained from a commercial culture and the species identification was confirmed. This dipteran is characterized by a limited flying ability. The insects had been reared in laboratory conditions for few generations. For the tests only adult individuals were used. Aphid *A. pisum* was obtained from the laboratory culture reared for many generations. Only adult wingless insects were used for the tests. Thirty aphids and 30 flies were placed separately on the bottom of glass vessels with plants, which then were tightly closed with a parafilm. The experiment was carried out at a temperature of 20 ± 1 °C with natural sunlight. The insects’ behaviour was observed during the first two and four hours of the experiment. After two and four days two randomly selected funnel-shaped leaves were cut along from each plant and their content was checked for insects (the pictures from the experiment are included in the Appendix A—Photographic documentation of the in vivo bioassays). The number of insects trapped was determined by subtracting both the living and dead specimens visible in the vessel from the initial number of insects.

#### 4.3.2. Bioassays with Pyridine

The bioassays were made using the olfactometry method with pyridine extracted from *Sarracenia alata* plants. Pyridine was selected because of its possible effect in attracting insects, specifically fruit flies [39]. The pyridine (Sigma Aldrich, Saint Louis, MO, USA, cat. No. 1.09728) effect on insects was tested in three concentrations [*v*/*v*]: 1 × 10^−3^, 1 × 10^−2^ and 1, with hexane as the solvent. The tests were performed using a four-choice olfactometer (at the University of Science and Technology in Bydgoszcz, Poland) with the test area 55 × 55 × 5.5 cm as described by Petterson [48] in room conditions (22 ± 2 °C). The air was removed from the center of the olfactometer by a vacuum pump, adjusted with a flow meter to 400 mL/min. The olfactometer arena was split into four areas: four by each way and additionally central area connected to pump for sucking air [49]. During the test one arm of the olfactometer was always assigned to the tested compound while three arms to the solvent (control). The tests were performed from the lowest to the highest concentration. Each odor chamber contained a filter paper disk (1.0 cm^2^) saturated with 50 μL of a solvent or corresponding solution.

The insects (the same species, as tested in bioassay) were introduced individually into the test area through the hole in the top of the olfactometer. Flies were able to make the choice within five minutes, while aphids were unable to effectively move in the test area. Therefore, only the response of flies was tested. During the test flies were introduced individually in 20 replications for each concentration. Each individual fly was exposed to a test concentration for seven minutes (two minutes to acclimatize in the olfactometer after which the experiment was run for further five minutes). The insects’ behaviour in the olfactometer was observed and the time of their first reaction measured. The choice of each fly was recorded. Particular individuals were used only once in the test.

The response was calculated as:(1)$$R=\frac{No.\;of\;individuals\;on\;the\;odour\;side-No.\;of\;individuals\;on\;the\;control\;side}{Total\;No.\;of\;individuals}$$

The data were analyzed in the SAS University Edition using the general linear model (GLM). The repeated factor was time (20 series for each concentration), while the area of the olfactometer (5 variants) was calculated as random effect.

#### 4.3.3. Statistical Analysis

The results were expressed as the mean of the measurements and reported as mean ± standard error. A two-way analysis of variance (ANOVA) was conducted to verify the lack of significance of PGRs (plant growth regulators), type of explant, and the interaction of explant type and the influence of PGR on *S. alata* micropropagation. The significant differences were assessed at levels of 0.05. When an analysis of variance gave a significant result, Tukey’s HSD test was performed to compare mean values.

## 5. Conclusions

Benzyl benzoate and pyridine, compounds which may play an important role in the attractant or repellent activity of plants, were identified in *Sarracenia alata* ex vitro and in vivo cultures. *Saracenia alata* in vitro cultures may be a promising source of sesquiterpeme compounds, namely β-caryophyllene, α-bergamotene and their derivatives such as caryophyllene oxide for instance, which displays a wide range of different bio-activity. Nevertheless, since the cultures of *Saracenia alata* were obtained in microscale, the further research which will allow to obtain plant material amount adequate for hydrodistillation, should be considered.

## Figures and Tables

**Figure 1 molecules-26-02406-f001:**
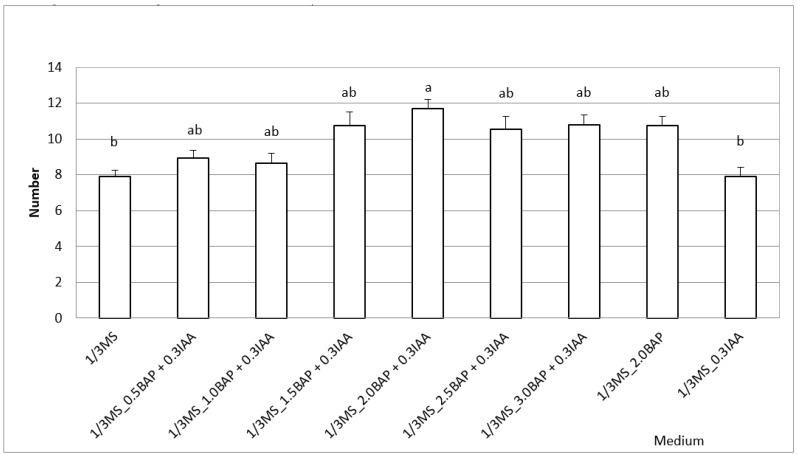
Summarized number of new shoots after two passages. Significance at *p* = 0.05. Values followed by the same letters are not statistically different in the Tukey test—one-way variance analysis.

**Figure 2 molecules-26-02406-f002:**
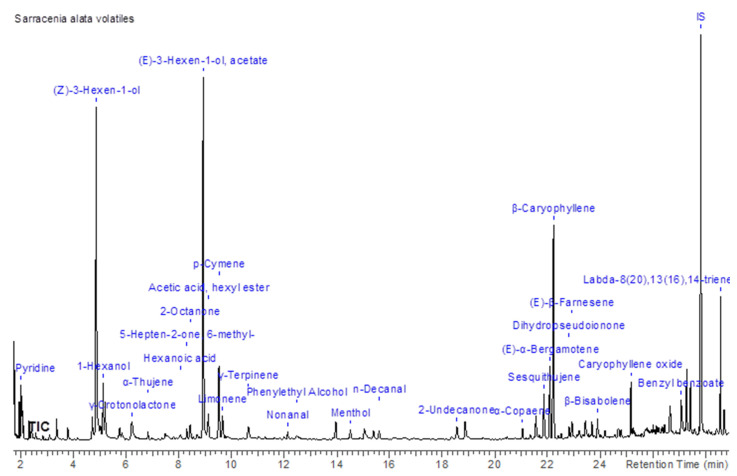
The volatile organic constituents’ profile of *Sarracenia alata* microcuttings.

**Figure 3 molecules-26-02406-f003:**
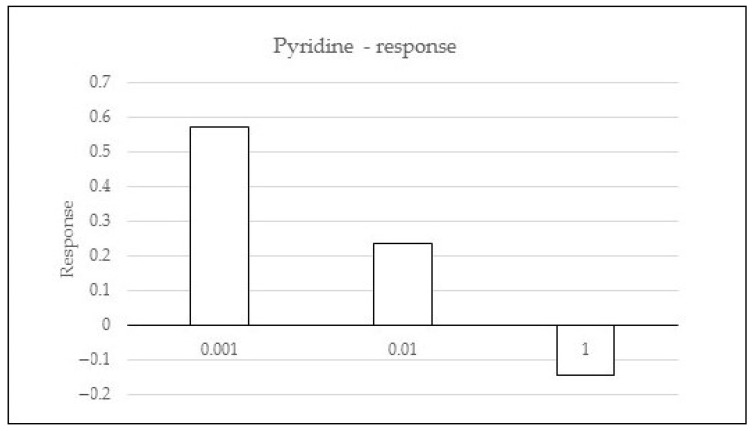
Response of *Drosophila hydei* to pyridine in three concentrations (%) as compared with the control.

**Table 1 molecules-26-02406-t001:** Influence of subsequent passages and medium composition on the development of *Sarracenia alata* microcuttings.

Medium	Height of Plants		No. of Leaves		No. of New Shoots		No. of Roots		Length of Roots		Fresh Weight		Dry Weight	
Summary influence on analysed traits after subsequent passages														
I passage	6.26 ± 0.10	a *	26.90 ± 0.82	a	5.81 ± 0.23	a	2.36 ± 0.17	a	0.99 ± 0.05	b	0.57 ± 0.02	a	0.07 ± 0.00	a
II passage	6.68 ± 0.15	b	22.30 ± 0.77	b	3.96 ± 0.16	b	2.16 ± 0.14	a	1.17 ± 0.05	a	0.52 ± 0.00	b	0.07 ± 0.00	a
Summary effect of medium after subsequent passages														
⅓MS	6.07 ± 0.34	ab	23.47 ± 1.78	a	4.08 ± 0.45	ab	3.58 ± 0.13	a	1.16 ± 0.11	a	0.31 ± 0.03	c	0.04 ± 0.00	c
⅓MS_0.5BAP + 0.3IAA	6.55 ± 0.40	ab	23.50 ± 1.37	a	4.47 ± 0.29	ab	1.70 ± 0.16	b	1.01 ± 0.09	a	0.38 ± 0.02	bc	0.05 ± 0.00	bc
⅓MS_1BAP + 0.3IAA	6.65 ± 0.33	ab	22.98 ± 1.47	a	4.32 ± 0.34	ab	1.99 ± 0.11	b	1.11 ± 0.10	a	0.56 ± 0.01	ab	0.07 ± 0.00	ab
⅓MS_1.5BAP + 0.3IAA	6.70 ± 0.23	ab	27.10 ± 1.92	a	5.37 ± 0.45	ab	1.74 ± 0.15	b	1.23 ± 0.09	a	0.65 ± 0.02	a	0.08 ± 0.00	a
⅓MS_2BAP + 0.3IAA	6.19 ± 0.19	ab	26.86 ± 2.15	a	5.84 ± 0.42	a	2.11 ± 0.11	b	1.04 ± 0.10	a	0.59 ± 0.04	a	0.08 ± 0.00	a
⅓MS_2.5BAP + 0.3IAA	7.12 ± 0.20	a	26.00 ± 0.98	a	5.26 ± 0.39	ab	1.79 ± 0.08	b	0.95 ± 0.08	a	0.71 ± 0.04	a	0.09 ± 0.00	a
⅓MS_3BAP + 0.3IAA	6.11 ± 0.20	ab	26.16 ± 1.83	a	5.47 ± 0.50	ab	1.57 ± 0.18	b	0.93 ± 0.00	a	0.66 ± 0.01	a	0.09 ± 0.00	a
⅓MS_2BAP	6.77 ± 0.21	ab	23.72 ± 1.46	a	5.37 ± 0.36	ab	1.72 ± 0.04	b	0.96 ± 0.06	a	0.64 ± 0.02	a	0.08 ± 0.00	a
⅓MS_0.3IAA	6.02 ± 0.23	b	22.10 ± 1.48	a	3.95 ± 0.35	b	4.20 ± 0.04	a	1.27 ± 0.10	a	0.37 ± 0.01	c	0.05 ± 0.00	c

* Values followed by the same letter within a column are not significantly different (*p* > 0.05, Tukey’s test)—part of two-way of variance analysis; ± SE—standard error.

**Table 2 molecules-26-02406-t002:** Number of insects found in the plant funnels after 2 and 4 days of incubation.

	Initial Number of Insects	After 2 Days of Incubation	After 4 Days of Incubation
*Drosophila hydei*	30	26	30
*Acyrthosiphon pisum*	50	10	10

## Data Availability

Data is contained within the article or Appendix A.

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
