# Peer review of "Sarracenia alata (Alph.Wood) Alph.Wood Microcuttings as a Source of Volatiles Potentially Responsible for Insects’ Respond"

_molecules, 2021, doi:10.3390/molecules26092406_

Round 1

Reviewer 1 Report

Dear editor

I write in reference to the manuscript “ Sarracenia alata Microcuttings as a Source of Volatiles Potentially Responsible for Insects’ Respond” by Łyczko and coworkers. The manuscript (Ms) refers to the micropropagation and of Sarracenia alata, its in vitro volatile organic compounds emission and its attractant effect over the insects Drosophila hydei and Acyrthosiphon pisum. The work also identifies the pyridine as an attractant of Drosophila hydei.

Although is not a completely novelty topic because there are previous works referring the effect of plant growth regulators on Sarracenia alata propagation and the metabolite profiling of the plant,the present MS is a nice piece of work focused in the identification of a metabolite (pyridine) with the potential to be used as attractant for a carnivore plant to caught its prey and in my opinion can be published after minor changes.

Minor changes:

Lines 20, 26 and 57. “in vitro” please check cursives.

Line 52. “[7]M” please correct.

Line 108. Please define BAP and IAA.

Table 1 and figure 1 should indicate the “n” size. Please use standard error instead standard deviation.

Why do the authors use p > 0.05 in table 1 but p = 0.001 in figure 1? Please homogenize.

Figure 2. Please use arrows to connect the peak with the compound name.

Figure 3. Please draw the Y axis.

Line 238. Was this study performed by the authors? Could the authors comment on the results?

M&M Please include the fabricant, city, country and catalogue number of the compounds.

Line 247. Please define KIN.

Line 250. Were the seeds germinated in the refrigerator? Were they incubed and germinated after 3-4 weeks? Please clarify.

Line 250. Please describe the explants.

Lines 248-255. I am a little confused. The seeds were germinated, then the explants (2.5 - 3 cm) were obtained, then the explants were cultivated with 1 KIN + 0.5 l 251 IAA and 1.5 BAP + 0.3 IAA for multiplication. Then secondary explants were obtained and cultivated two passages without PGRs, and then they were used in the experiment reported in table 1. That is correct? Could the author add a diagram as supplementary figure?

Line 256. 2.5-3.0?

Line 270. This seems to be the description of a ex vivo determination. Was the plant material a living plant? Was the living plant in culture media or soil/substrate? Please clarify. Depending on the answer the VOCs determination was ex vivo, in vitro or in vivo and according with this could be necessary to adjust the abstract (lines 20 and 21) , introduction (line 92) and conclusions (lines 345 and 346). 

Line 315. Was the concentration calculated as V/V? Please clarify. 

About the olfactometer. Can the authors provide an olfactometer photograph as supplementary figure?

Linne 324. Please provide the volumes of the solvent (control) and the pyridine solution. 

Figure S7. Can the authors point to the digested flies bodies?

Author Response

Dear Reviewer,

Thank you for your valuable comments regarding manuscript molecules-1158213, entitled Sarracenia alata Microcuttings as a Source of Volatiles Potentially Responsible for Insects’ Respond. We appreciate your detailed review and hope that our corrections will find your acceptance.

Reviewer 2 Report

I have reviewed the manuscript: molecules-1158213, titled:  Sarracenia alata Microcuttings as a Source of Volatiles Potentially Responsible for Insects’ Respond.

The argument is not well presented and the content of the manuscript is not really consistent with the aim of the study indicated in the introduction section. The experiment is not well described and presented. The Authors want to investigate the response of insects to the exposure to Sarracenia plants but they did not include the percentage composition of volatiles and the volatile composition is not compared the with the one of other species. In addition, the argument of this manuscript is not really interesting for a Reader of Molecules. In view of these criticisms I suggest the rejection of this manuscript.

Regards.

Author Response

Dear Reviewer,

Thank you for your valuable comments regarding manuscript molecules-1158213, entitled Sarracenia alata Microcuttings as a Source of Volatiles Potentially Responsible for Insects’ Respond. Thank you for your time, which was spent for evaluation of our manuscript.

Some literature sources were updated due to better argumentation of the study.  We do not agree that the experience will not be of interest to the reader. We have done the possible, at this stage, research and found the results very interesting. Therefore, we would like to share them with the readers.

 We will not be able now to change the fact that there was one plant, moreover it was not the intention of the research and we do not know the percentage of the substance, but the data for the tests are as accurate as possible and, in our opinion, sufficient for a correct interpretation. The identification of substances responsible for the action on insects is a significant and new achievement resulting from this work.

       Sincerely,

Jacek Łyczko, corresponding author.

Reviewer 3 Report

In this manuscript, the authors have reported chemical composition of volatiles secreted from carnivorous plant (Sarracenia alata) responsible for insects’ response.

Here are my comments,

1) Page 1, Lines 40-42. the sentence is confusing to the readers. Please rephrase it.

2) Introduce and elaborate the acronym IUCN, before using the abbreviation. (Page 2, Line 47).

3) Page 2, Line 51. There is typo with extra M after [7]. Please correct.

4) Page 2, Line 57. Please Italicize In vitro.

5) Page 2, Lines 64-65. the sentence is confusing to the readers. Please rephrase it.

6) Introduce and elaborate the acronyms for media BAP, IAA, KIN, and MS before using the abbreviation. (Page 3, 2nd paragraph).

7) Also, introduce and elaborate the acronyms DVB/CAR/PDMS SPME, NIST, LRI, and PPTE before using the abbreviation. (Page 8, Lines 277 - 279).

8) Page 9, Lines 335-338. Please break the sentences into individual words in the equation. (e.g. Numberofindividualsontheodourside to Number of individuals on the odor side or # of individuals on the odor side).

9) Mention name of the plant in SI (Figure S1).

10) Typo in Figure S3 (SI), four davs instead of four days. I think, the Figure S4 is overlapping the text of Figure S3.

To conclude, given the originality, and interest to the scientific community I am happy to recommend the editors to accept the manuscript to publish in Molecules with minor changes as mentioned in my comments 1 to 10.

Author Response

(The authors gave the same response as above.)

Reviewer 4 Report

Dear Authors,

After the review process, I have several comments: you should insert numerical data in the abstract; you should include more new data in the introduction, based on new references from the last five years. For example, you should mention studies about the bioactive potential of functional products and the bioavailability of phenolic compounds, as secondary metabolites; you should include a statistical section at the end of Materials and Methods; you should include references in all Materials and Methods sections; you should present limitations for the study and future valorization.

Best regards!

Author Response

(The authors gave the same response as above.)

Round 2

Reviewer 2 Report

To the Authors (in detail)

  • In the title and in the whole manuscript: To avoid confusion, please use the correct and updated botanical nomenclature, for example according to www.gbif.org, and also report the authorship and (in brackets) the botanical family in the title and at the first mention in the text. However, you could report the scientific name in the Abstract just like Sarracenia alata. In the rest of the text it is possible to indicate the species as alata.

  • For all the other scientific names use the International rules. When you write the scientific name for the first time also report the authorship and (in brackets) the botanical family. In the rest of the text it is possible to indicate the species as alata;

  • In the tiles and sub-titles of the manuscript, when you indicate the scientific names, do not include the authorship and the botanical family, only the genus and the specie;

  • Abstract section, include some relevant numeric value of the volatile percentage composition of alata;

  • Line 41 and in the whole manuscript: the scientific names in italics;

  • Line 76 and throughout the entire manuscript: when you write the bibliography, apply the guidelines of Molecules. In this case, delete the year of publication;

  • Discussion section. This section has to be improved and your data have to be compared with findings of other authors to discuss similarity of the volatile composition of alata with the volatile composition of other species. In particular, you have found: α-Thujene, p-Cymene, γ-Terpinene, Nonanal, Copaene, α-cis-Bergamotene, (E)-β-Farnesene, Caryophyllene oxide which are also contained in the volatile composition other species [X1, X2]. In particular, bergamotene is contained in the bergamot peel essential oil: Citrus bergamia Risso (Rutaceae) [X2]. Here I have listed two very recent scientific works to be find, read and discussed in your manuscript, but you have to find, read and discuss some paper more in relation with the volatile composition of S. alata.  

.

[X1] Shebaby W.; Saliba J.; Faour W.H.; Ismail J.; El Hage M.; Daher C.F.; Taleb R.I..; Nehmeh B.; Dagher C.; Chrabieh E.; Mroueh M.

In vivo and in vitro anti-inflammatory activity evaluation of Lebanese Cannabis sativa L. ssp. indica (Lam.)

 Journal of Ethnopharmacology, 270 (2021) Article number 113743.

[X2] Gioffrè, G.; Ursino, D.; Labate, M.L.C.; Giuffrè, A.M.

The peel essential oil composition of bergamot fruit (Citrus bergamia, Risso) of Reggio Calabria (Italy): a review.

Emirates Journal of Food and Agriculture 32(11): 835-845 (2020)

doi: 10.9755/ejfa.2020.v32.i11.2197

  • Line 157, write well the scientific name: alata;

  • Lines 188, 195, 208, 211, 295, 345 and in the whole manuscript: when you write the bibliography, apply the guidelines of Molecules. In this case, delete the year of publication;

  • Line 201 mg/l, lines 297 and 347 ml, line 352 mL: when you indicate the liter, sometime you use the small and sometime the capital letter, please be consistent in the whole manuscript;

  • Line 248 and in the whole manuscript, you have to be more clear for the symbol you have used: if you are meaning microliters the symbol is: μg and not ug. You have to use the font of the ancient Greek alphabet;

  • Lines 262 and 347 and in the whole manuscript: please be consistent in the whole manuscript when you indicate the unity of measurement: Y/N for the dot between the mg and mL or between mg and dm?

  • Line 306 and in the whole manuscript, when you write the temperature, separate the value from the symbol: 50 °C and not 50°C;

  • Table S2 include one column or create a new column with the volatile percentage composition alata ±SD and discuss your findings in the discussion section;

  • Throughout the entire manuscript and in the tables and figures, specify if it is α or β-Copaene;

  • In the whole manuscript and tables and figures, specify if it is β-Caryophyllene;
  •  
  • In the manuscript it is not really evident if you have written (Farnesene r n) or Famesene m. The correct name is Farnesene r n, please, verify;
  •  
  • Please, write in blue color (anyway differently evidence) the corrections you will do and make carefully all the listed corrections.

Author Response

Dear Reviewer,

Thank you for your valuable comments regarding manuscript molecules-1158213, entitled Sarracenia alata (Alph.Wood) Alph.Wood Microcuttings as a Source of Volatiles Potentially Responsible for Insects’ Respond. We appreciate your detailed review and hope that our corrections will find your acceptance. Please not, that all improvements and correction done during 2nd review round, were pointed out by blue font color or highlight by blue color.

  • In the title and in the whole manuscript: To avoid confusion, please use the correct and updated botanical nomenclature, for example according to www.gbif.org, and also report the authorship and (in brackets) the botanical family in the title and at the first mention in the text. However, you could report the scientific name in the Abstract just like Sarracenia alata. In the rest of the text it is possible to indicate the species as alata.
  • For all the other scientific names use the International rules. When you write the scientific name for the first time also report the authorship and (in brackets) the botanical family. In the rest of the text it is possible to indicate the species as alata;
  • In the tiles and sub-titles of the manuscript, when you indicate the scientific names, do not include the authorship and the botanical family, only the genus and the specie;

 The Reviewer suggestions, regarding botanical nomenclature and proper scientific names has been included. The corrections were introduce into the manuscript.

  • Abstract section, include some relevant numeric value of the volatile percentage composition of alata;

 The information regarding major volatile compounds percentage has been included in Abstract section.

  • Line 41 and in the whole manuscript: the scientific names in italics;

 It has been corrected.

  • Line 76 and throughout the entire manuscript: when you write the bibliography, apply the guidelines of Molecules. In this case, delete the year of publication;

 The year of publication has been deleted throughout the entire manuscript.

  • Discussion section. This section has to be improved and your data have to be compared with findings of other authors to discuss similarity of the volatile composition of alata with the volatile composition of other species. In particular, you have found: α-Thujene, p-Cymene, γ-Terpinene, Nonanal, Copaene, α-cis-Bergamotene, (E)-β-Farnesene, Caryophyllene oxide which are also contained in the volatile composition other species [X1, X2]. In particular, bergamotene is contained in the bergamot peel essential oil: Citrus bergamia Risso (Rutaceae) [X2]. Here I have listed two very recent scientific works to be find, read and discussed in your manuscript, but you have to find, read and discuss some paper more in relation with the volatile composition of S. alata.  

The mentioned references, and other ones, has been included in Discussion section. Please see lines 224-233. 

  • Line 157, write well the scientific name: alata;

 It has been added.

  • Lines 188, 195, 208, 211, 295, 345 and in the whole manuscript: when you write the bibliography, apply the guidelines of Molecules. In this case, delete the year of publication;

 It has been done.

  • Line 201 mg/l, lines 297 and 347 ml, line 352 mL: when you indicate the liter, sometime you use the small and sometime the capital letter, please be consistent in the whole manuscript;

 It has been corrected.

  • Line 248 and in the whole manuscript, you have to be more clear for the symbol you have used: if you are meaning microliters the symbol is: μg and not ug. You have to use the font of the ancient Greek alphabet;

 It has been corrected.

  • Lines 262 and 347 and in the whole manuscript: please be consistent in the whole manuscript when you indicate the unity of measurement: Y/N for the dot between the mg and mL or between mg and dm?

 It has been corrected.

  • Line 306 and in the whole manuscript, when you write the temperature, separate the value from the symbol: 50 °C and not 50°C;

 It has been corrected

  • Table S2 include one column or create a new column with the volatile percentage composition alata ±SD and discuss your findings in the discussion section;

The column has been added and the findings were discussed.

  • Throughout the entire manuscript and in the tables and figures, specify if it is α or β-Copaene;
  • In the whole manuscript and tables and figures, specify if it is β-Caryophyllene;
  • In the manuscript it is not really evident if you have written (Farnesene r n) or Famesene m. The correct name is Farnesene r n, please, verify;

The mentioned volatile compound names were verified and corrected in Table S2, Figure 2 and relevant places in the text.

Reviewer 4 Report

DearAuthors,

You should improve the presentation of the introduction section. You should include new methods, innovative, to determine the bioavailability of the phenolic compound related to their bioactivity. You should read very carefully my first review. You mention only two studies, but no mechanism and methods (e.g., in vivo lab method).

Best regards,

Author Response

Dear Reviewer,

Thank you for your valuable comments regarding manuscript molecules-1158213, entitled Sarracenia alata (Alph.Wood) Alph.Wood Microcuttings as a Source of Volatiles Potentially Responsible for Insects’ Respond. We appreciate your detailed review and hope that our corrections will find your acceptance. Please not, that all improvements and correction done during 2nd review round, were pointed out by blue font color or highlight by blue color.

You should improve the presentation of the introduction section. You should include new methods, innovative, to determine the bioavailability of the phenolic compound related to their bioactivity. You should read very carefully my first review. You mention only two studies, but no mechanism and methods (e.g., in vivo lab method).

The relevant information was introduced into Introduction section. Please see lines 78-89. We have found several (3) references which provided comprehensive information regarding bioactive compounds and their bioavailability.